# Experimental evidence that penis size, height, and body shape influence assessment of male sexual attractiveness and fighting ability in humans

Upama Aich[1,2,3]*, Chloe Tan[3], Rebecca Bathgate[3], Khandis R. Blake[4], Robert C. S. Capp[4], Jacob C. Kuek[4], Bob B. M. Wong[2], Brian S. Mautz[3], Michael D. Jennions[3,5]

1 Centre for Evolutionary Biology, School of Biological Sciences (M092), University of Western Australia, Crawley, Western Australia, Australia, 2 School of Biological Sciences, Monash University, Clayton, Victoria, Australia, 3 Division of Ecology and Evolution, Research School of Biology, The Australian National University, Canberra, Australian Capital Territory, Australia, 4 Melbourne School of Psychological Sciences, The University of Melbourne, Melbourne, Victoria, Australia, 5 Stellenbosch Institute for Advanced Study (STIAS), Wallenberg Research Centre at Stellenbosch University, Stellenbosch, South Africa

* aich.aich49@gmail.com

## Abstract

Why the human penis is unusually large compared to that of other primates is a long-standing evolutionary question. Sexual selection, through female mate choice and male-male competition, is a likely driver, but confirming this is difficult due to natural covariation among traits. The solution is to experimentally manipulate focal traits to identify targets of selection. Using 343 computer-generated male figures that varied in penis size, height and body shape, we experimentally tested how these traits influence perceived attractiveness and fighting ability. Over 800 participants—both male and female—viewed either life-sized (in-person) or scaled (online) animations and rated the figures. Across both settings, selection analyses revealed consistent directional selection favoring taller men with a more V-shaped body and a larger penis. In both surveys, male participants rated rivals with a larger penis as more sexually competitive and physically threatening. To our knowledge, this is the first experimental evidence that males assess rivals' fighting ability and attractiveness to females based partly on a rival's penis size. Our findings suggest that female choice and male-male competition have jointly favored larger penis size, greater height, and more V-shaped bodies in men.

## Introduction

Among primates, the human penis is an evolutionary outlier [1]. Controlling for body size, the human penis has a greater girth and is generally longer than that of other

---

**Data availability statement:** All relevant data and code are within the paper and added as Supporting information files.

**Funding:** The following funding sources supported this work: U.A. received funding from the Monash University Research Reactivation Grant and the Forrest Research Foundation Fellowship (2023/GR001415) (https://www. forrestresearch.org.au). B.S.M. was supported by the National Cancer Institute (grant T32 CA160056) (https://www.cancer.gov). B.B.M.W. received funding from the Australian Research Council (grants DP220100245 and DP250100501) (https://www.arc.gov.au). M.D.J. received funding from the Australian Research Council (grant DP2019100279) (https://www. arc.gov.au). The funders had no role in study design, data collection and analysis, decision to publish, or preparation of the manuscript.

**Competing interests:** The authors have declared that no competing interests exist.

**Abbreviation:** LRT, likelihood ratio test.

primates [1,2]. Unlike the penis of other great apes, the human penis also lacks a baculum (penis bone) and relies solely on blood flow during erection to achieve rigidity. Prior to clothing, the human penis was a visually prominent character, akin in its prominence to that of the enlarged mammary glands of women. Such exaggerated sexually dimorphic traits always generate speculation about the roles that sexual and natural selection have played in their evolution [3–5]. For example, it has been suggested that coevolution in response to the wide birth canal of women has selected for penis shape and size in humans [6,7]. In general, however, far more scientific research has asked why humans have enlarged mammary glands than has asked about the size of the human penis [1,8]. In the current study we therefore test two potential explanations that invoke sexual selection to explain the enlarged human penis.

Sexual selection has been identified as a key driver of male genital evolution across most taxa in which it has been studied [9]. For example, there is far less variation among species in male genital morphology in taxa where females are monogamous than in taxa where females mate with multiple males [10]. Sexual selection is equally likely to have affected the evolution of female genitalia and reproductive tracts due to male–female coevolution, but there are far fewer studies documenting female genital diversity [11], but see: [12,13]. To date, most research has focused on how post-copulatory sexual selection affects the evolution of male genitalia. Specifically, researchers have asked how the size, shape, and fine-scale structure of male genitalia increase sperm competitiveness by increasing the removal of rivals' sperm and/or improving the retention and utilization of a male's own sperm for later egg fertilization [6,9,14]. There is also good evidence that cryptic female choice has favored specific features of male genitalia that increase the likelihood a female will use a male's sperm for fertilization [15]. In the case of humans, some studies using artificial models have led to the hypothesis that the enlarged size of the penis and the shape of the glans act to increase sperm displacement [6]; and that a wider penis is associated with greater physical stimulation, which increases the likelihood of female orgasm that then improves sperm retention [16–18]. However, other research suggests that human sperm competition strategies may be primarily defensive (e.g., related to ejaculate volume) rather than offensive via physical displacement, questioning the strength of this selective pressure [19].

In contrast to the many studies on post-copulatory sexual selection, few studies have tested how pre-copulatory sexual selection—which favors traits that increase the likelihood of mating—has affected the evolution of male genitalia. This research bias is understandable: in most taxa, male genitalia are hidden inside the body and are therefore invisible to females (e.g., most insects). Male genitalia are, however, readily apparent in some species, such as poecilid fish, some mammals and many primates, including humans.

In animals, there are two main mechanisms of pre-copulatory sexual selection on males [3,14]. First, female mate choice can favor the exaggeration of traits that make a male more sexually attractive. These traits are often labeled as ornaments, sexual advertisements, or sexual signals. Second, male-male competition can favor traits that increase a male's likelihood of success during a physical contest. These

traits include weaponry (e.g., horns, antlers, and tusks), increased musculature, and greater body size. Crucially, however, male-male competition can also favor the evolution of signals that indicate a male's fighting ability and thereby reduce his likelihood of being challenged by an inferior rival (e.g., badges of status or dominance) [20]. For example, larger coloured patches have been shown to lower the risk that a male is attacked by his rivals in birds [21], lizards [22], and damselflies [23]. In humans, men strategically pick the targets of their aggression [24] and tend to consider taller men with broad shoulders as more aggressive and skilled fighters [25]. It should be noted that some male traits appear to increase both attractiveness to females and signal fighting ability [26]. In humans, for example, traits linked to greater fighting ability, such as greater height and broader shoulders, also increase male attractiveness to the opposite sex [27–30].

Pre-copulatory sexual selection on male genitalia has only been examined in a few studies. To do so convincingly, it is necessary to manipulate a focal trait to show that it is under direct sexual selection, thereby eliminating natural correlations with other traits that might be the true target of selection. There are only a handful of such experiments. Experimental studies of two species of poecilid fish have manipulated the length of the male intromittent organ, which is a modified anal fin used to transfer sperm, to show that a longer fin is more attractive to females [31,32]. In humans, a few experimental studies have presented women with computer-generated images or drawn figures to test if penis size affects male sexual attractiveness [33–36]. These studies report that females find a larger penis more attractive [33,34]. However, the strength of the effect is moderated by male height and body shape, with penis size having a greater effect on the attractiveness of taller men and men with a more V-shaped body [35,37].

While there is little evidence that male genitals have evolved as weapons under pre-copulatory sexual selection [38], it is certainly plausible that male genitalia could function as status signals that affect the likelihood of rivals initiating a fight [39,40]. For example, genital size could reflect the circulating levels of hormones that affect the development of primary and secondary sexual characters, as well as traits that increase fighting ability (e.g., musculature or aggressiveness). In mammals, testosterone affects genital development, musculature, and aggressiveness, thereby providing a plausible link between penis size and fighting ability [41,42]. In humans, it has been speculated that a larger penis is viewed as a signal of social dominance and might therefore influence male assessment of a rival's fighting ability [43]. To date, however, no study in any animal, including humans, has experimentally modified male genital size to test whether it acts as a signal of fighting ability.

In the current study, we use 343 computer-generated figures of men created by [35] (**Fig 1**) to test how male genital size influences the assessment of a male's attractiveness by females, and how males perceive a rival male's attractiveness to females and evaluate his fighting ability. Specifically, 600 male and 200 female participants rated figures that independently varied in three traits: height, body shape, and penis size (7 values per trait; $7^3 = 343$). Females scored the figures for sexual attractiveness, while males scored the figures for either fighting ability or the threat posed as a sexual rival. We then tested how the three traits interacted to affect: (a) a female's assessment of sexual attractiveness; (b) a male's assessment of fighting ability; and (c) a male's assessment of the level of sexual competition, which we assume is related to the perceived likelihood (threat) that a rival will mate with the participant's sexual partner. We used standard evolutionary selection analyses to calculate linear, quadratic, and correlational selection on the three traits using the ranking score as a measure of fitness [35]. We also tested if the method used to view the figures affected the results by comparing selection estimates when life-sized figures were presented to estimates generated when figures were presented online (i.e., when viewed as smaller images on a laptop, desktop phone or tablet). We predicted that larger penis size, greater height, and a larger shoulder-to-hip ratio would all increase male attractiveness and perceived fighting ability. We also predicted these effects would be greater when participants viewed life-sized figures.

## Materials and methods

### Test figures and videos

We created 343 computer-generated, anatomically accurate male figures in MakeHuman (v0.9.1RC1) as part of a previously published study [35]. Briefly, the figures only varied in three traits: flaccid penis size, height, and body shape

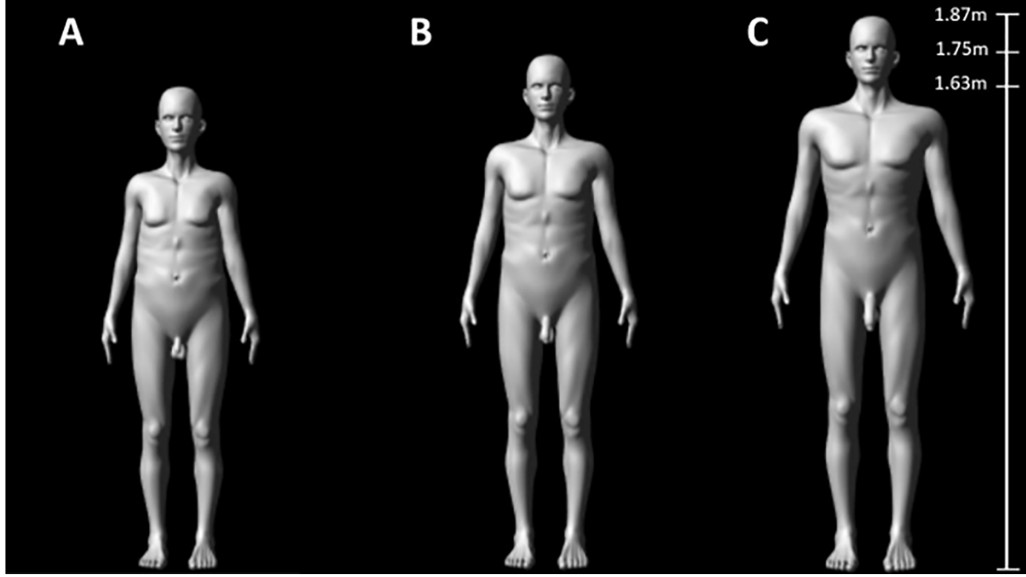

**Fig 1. Examples of the computer-generated, male figures used in our study.** Figure **(A)** has the smallest values for penis size, height, and body shape (shoulder-to-hip ratio), Figure **(B)** has average values for each trait, and Figure **(C)** has the largest values for each trait [35].

(shoulder-to-hip ratio). Each trait was represented by seven values spanning a natural range (penis size: 5–13 cm, height: 1.63–1.87 m, body shape: 1.13–1.45; i.e., pear to V-shaped). Longer penises were also wider, with a 1.2 cm increase in width between the shortest and longest penis, and we therefore refer to the trait as "penis size." The 7 values for height and penis size were evenly distributed across a range spanning ±2 SD of the mean of a sample of 3,300 Italian men that captures ~95% of the variation that humans are likely to encounter [44]. Body shape values were evenly spaced between 1.13 and 1.45, with a mean and SD of 1.274 ± 0.107 ranging from pear to V-shaped. These values fell within the natural range of a sample of White men [45]. The 343 figures represent all possible combinations of the trait values (7 × 7 × 7).

Each figure was presented as a video in which the figure rotated 30° to the left and then 30° to the right over 4 s. Rotation increased the ability of participants to gauge penis size and body shape. We created 100 unique sets of 56 figures: 52 test figures and 4 control figures. The control figure had the mean values for all three traits, and was presented as the 1st, 15th, 29th, and 43rd figure to ensure the participants regularly saw a benchmark figure with the average values. Each test figure (excluding the control) was in 15 or 16 of these 100 sets (5,200/342 = 15.2) and presented in a random place in the order sequence. In total, each of the 343 figures was rated by 12–18 participants (an average of 15 ratings per figure).

## Participants

We ran both in-person and online surveys to test if the mode of delivery affected the results. In both cases, there were three types of surveys: females rating male attractiveness, males rating a rival male's attractiveness, and males rating a rival male's fighting ability. We collected data for the in-person surveys using PsychoPy3 [46]. We recruited participants using posters and targeted media posts at the Australian National University (ANU), Monash University, and the University of Melbourne and incentivised participation with the chance to win one of two $100 lottery prizes. We briefly described the project as examining physical predictors of male attractiveness or fighting ability, but participants were not told which male traits varied.

For in-person surveys, participants were instructed to stand 6.5m in front of life-sized figures that were projected one at a time onto a white wall. Each participant was alone in the test room while the researcher waited outside. Participants began by sequentially rating a set of 13 training figures that spanned the full range of trait values. This allowed participants to familiarize themselves with the videos and the rating procedure. They then proceeded to rate the 56 figures (52 test, 4 control) in their assigned set. Rating was on a 7-point Likert scale. Female participants were asked to rate figures based on the question: "How sexually attractive is this figure to you?" (1 = very unattractive, 7 = very attractive). To determine how male participants estimated the attractiveness of these figures to females, they were asked to rate figures based on the question: "How jealous would you feel if you saw this man talking to your partner?" (1 = not jealous at all, 7 = extremely jealous). We assume that increased jealousy is related to a rival being perceived to be more sexually competitive. To determine how male participants assessed the fighting ability of these figures, they were asked the question: "How threatened would you feel if this man wanted to pick a fight with you?" (1 = not threatened at all, 7 = extremely threatened). Each participant was only asked one question. The participant entered a 1–7 rating on a keyboard on a lectern next to them, and the next figure in the sequence then immediately appeared. The system automatically recorded the time between a figure appearing and a rating score being entered. Upon completion of the rating process, participants filled in a brief questionnaire about their own physical characteristics (e.g., height, weight, and age) and sexuality (four choices: heterosexual/bisexual/homosexual/other). Measurement tools (weighing scale and tape measure) were provided in the testing room.

We ran online surveys using PsyToolkit [47], which allowed participants to view and rate the videos remotely on their own devices. Videos were converted to GIF format for compatibility reasons. As with the in-person surveys, participants were randomly assigned one of the 100 unique sets of figures. They completed a questionnaire before sequentially viewing the 13 training figures and then viewed the 56 figures in the test set, with each figure presented one at a time. This procedure is identical to the procedure for in-person surveys. Online surveys were completed by participants who were either paid or unpaid. For paid surveys, participants were recruited via Prolific, a vetted platform, and compensated £1.70 for participation. Unpaid surveys were advertised through social media networks targeting eligible participants.

In total, we recruited over 600 men and over 200 women (approximately 100 for each of the 8 surveys). In addition, we incorporated data from a previous study [35] into our analysis, in which we ran an in-person assessment of male attractiveness by female participants ($n = 105$) using the same stimuli that we used in the current study. Inclusion of the data from the past study [35] provided us with the opportunity to directly compare the relative strength of intersexual selection (female choice) and intrasexual selection (male–male assessment) using identical stimuli for both in-person and online surveys. A minor error (1 of 5,145 points) in data entry was noted, so the results differed very slightly from the originally reported paper. We restricted data analysis to participants who self-identified as heterosexual; and we excluded any participants who stated that they were familiar with the study of [35]. The final sample sizes for each of the 9 surveys are presented in Table A in S1 Text (range: $n = 89$–105).

We compared the paid and unpaid online surveys and found that there was a significant effect of payment on the mean scores (i.e., the intercept; Table B in S1 Text). Crucially, however, the patterns of selection (i.e., the slopes for how each male trait affected a figure's rating) were qualitatively very similar for the two groups (see Tables A1-D in S1 Appendix). Given this consistency in the key findings, we chose to present only the unpaid sample in the main text. This simplifies the analysis and reduces any potential risk that some of the paid participants were less attentive to the survey figures or completed the survey only for financial gain.

## Ethics statement

Ethics approval was granted by the Australian National University (2020/084), Monash University (32191), and University of Melbourne (25368) Human Research Ethics Committees. Participation was voluntary, with the option to withdraw at any time. Data collection was anonymous: no answers could be traced back to individual participants.

## Statistical analysis

All analyses were run in RStudio version 4.4.3 [48]. Data on ratings for attractiveness and fighting ability were analyzed using standard multivariate selection procedures. To test the effect of penis size, height, and body shape (shoulder-to-hip ratio) on attractiveness and fighting ability, we used a standard analysis based on a multiple regression of "relative rating" (attractiveness or fighting ability) on standardized trait values (mean = 0, SD = 1). We centered the rating scores from each participant (i.e., the mean rating for each participant was zero). We centered the rating scores to correct for variation among participants in their tendency to consistently give higher or lower than average scores. For the relative rating of each of the 343 figures, we then calculated its mean participant-corrected attractiveness score (12–18 participant ratings per figure). The mean relative rating score of the 343 figures is 0, so we added 1 to generate a final relative rating score that is a surrogate of fitness, which traditionally has a mean value of 1. We estimated selection gradients (linear, quadratic, and correlational) and associated *P* values from standard tests for regression coefficients (see **Table 1**). We present the results as a selection analysis so the regression coefficients for the squared product of individual traits are doubled [35,49]. Selection gradients in Table 1 can be read as the change in rating score (on the original 1–7 scale) with a one SD increase in the focal trait.

For each of the three types of survey, we used a likelihood ratio test (LRT) to compare a model with or without the mode of delivery (in-person or online) included as a fixed factor to test if it influenced net selection on the three focal traits (i.e., the shape of the selection surface). There was an effect (see Results). We, therefore, ran separate models for in-person and online surveys. For each mode of delivery, we then used LRTs to determine: (a) whether the sex of the participants influenced how the three focal traits affected their assessment of attractiveness; (b) whether males differed in how the three focal traits affected their assessment of attractiveness and fighting ability; (c) whether there was a difference in how the three focal traits affected female assessment of attractiveness and male assessment of fighting ability. In all cases, there was always an effect (Table B in S1 Text), so we ran six separate models based on the mode of delivery, the sex of the participant and the question being asked.

**Table 1. Linear selection gradients (β) and the matrix (γ) of quadratic (on diagonal) and correlational (below diagonal) selection gradients using the average rating for each of the 343 figures. Results are for A in-person and B online surveys to measure: (a) female rating of male attractiveness, (b) male rating of other males' attractiveness, and (c) male rating of fighting ability.**

| Trait | Linear (β) | | Quadratic (γ) | | | | | |
| --- | --- | --- | --- | --- | --- | --- | --- | --- |
| | | | Penis size | | Height | | Body shape | |
| | A | B | A | B | A | B | A | B |
| **(a) Female rating of male attractiveness** | | | | | | | | |
| Penis size | **0.246*** | **0.517*** | **−0.10*** | **−0.19*** | | | | |
| Height | **0.269*** | **0.114*** | **0.062** | 0.03 | **−0.11*** | −0.042 | | |
| Body shape | **1.06*** | **0.768*** | **0.07*** | **0.117*** | **0.084*** | **0.059*** | **−0.288*** | **−0.216*** |
| **(b) Male rating of a rival's attractiveness** | | | | | | | | |
| Penis size | **0.307*** | **0.208*** | 0.01 | 0.042 | | | | |
| Height | **0.212*** | **0.068*** | 0.017 | −0.013 | 0 | −0.01 | | |
| Body shape | **0.46*** | **0.313*** | **0.035** | **0.027*** | **0.065*** | **0.025*** | −0.05 | −0.036 |
| **(c) Male rating of a rival's fighting ability** | | | | | | | | |
| Penis size | **0.061*** | **0.069*** | 0.034 | 0.01 | | | | |
| Height | **0.486*** | **0.201*** | 0.013 | −0.006 | **0.092** | 0.01 | | |
| Body shape | **0.312*** | **0.278*** | −0.006 | 0.015 | **0.028*** | **0.022*** | −0.014 | −0.018 |

*P* values are from the coefficients, and SE estimates from the multiple regression analysis. The dependent variable is the relative rating score per figure (see text for details). Bold values indicate statistical significance. Asterisks indicate FDR rate at \*\*\**P* < 0.001, \*\**P* < 0.01, \**P* < 0.05 (see Methods).

We also used the multiple-regression approach to calculate the fitness surface for each participant. We did so as our first analysis does not fully account for participant identity. In this second analysis, the dependent variable was the centered ranks for each participant, while the three focal traits were each standardized (mean = 0, SD = 1) for the unique set of 52 test figures that the participant rated. We then calculated the mean value across participants for each linear, quadratic, and correlational selection gradient. Each mean is based on one value per participant. We used one-sample *t*-tests to test if the mean selection gradient differed from zero (Table C in S1 Text). The first (one rank per figure) and second (one participant per selection coefficient estimate) analyses produced highly congruent findings (see Results).

To investigate sources of variation among participants in how penis size, height, and the shoulder-to-hip ratio affected their assessment of attractiveness or fighting ability, we calculated the Pearson's correlation between a participant's linear selection gradients for each trait and his/her age, height, and body mass index (was calculated as the residual of regression of weight on height for each sex). There are nine correlations per survey (**Table 2**).

Finally, we tested if penis size affected the time taken to rank a figure. We ran a general linear mixed model with log-transformed response time as the dependent variable and the three standardized male traits as fixed factors. We included participant identity as a random factor because there were multiple trials per participant. We excluded response times longer than 20 s (<2% of the total ratings), as this was a natural 'break' in the distribution of response times. However, the analysis yielded near identical conclusions if we used all the response times (Tables 3 and D in S1 Text).

We accounted for multiple testing for the full set of p-values obtained from testing the effect of penis size, height, and body shape, their interactions, the unpaid mode of delivery (in-person or online), participant sex, and study question (attractiveness or fighting ability) using the False Discovery Rate method [50]. We present original P values in text and tables, and we use asterisks to indicate whether the FDR threshold is ***$P<0.001$, **$P<0.01$, or *$P<0.05$. The original and adjusted *P*-values are included as a supplementary file in S1 Data.

## Results

Given the highly significant influence of the mode of delivery, sex of the participant, and the type of assessment (attractiveness or fighting ability) (Table B in S1 Text), we ran six separate models.

**Table 2. Correlations (*r*) between participant traits and the strength of linear selection (β) on figure traits for in-person and online surveys for: (a) female rating of male attractiveness; (b) male rating of male attractiveness; and (c) male rating of fighting ability.**

| | In-person participant traits | | | Online participant traits | | |
|---|---|---|---|---|---|---|
| *Male trait* | Age | Height | Relative weight | Age | Height | Relative weight |
| **(a) Female rating of male attractiveness** | | | | | | |
| Penis size | 0.008 | −0.088 | **0.329**\*\* | 0.030 | 0.019 | −0.077 |
| Height | −0.089 | **0.321**\*\*\* | 0.043 | −0.038 | **0.269**\* | 0.096 |
| Body shape | −0.159 | −0.064 | −0.062 | −0.101 | 0.043 | 0.008 |
| **(b) Male rating of a rival's attractiveness** | | | | | | |
| Penis size | **0.322**\*\* | −0.149 | 0.082 | −0.093 | −0.030 | −0.005 |
| Height | −0.202 | **0.402**\*\*\* | −0.053 | −0.229 | 0.101 | −0.191 |
| Body shape | **−0.2975**\* | 0.137 | −0.207 | **−0.301**\*\* | 0.243 | −0.128 |
| **(c) Male rating of a rival's fighting ability** | | | | | | |
| Penis size | −0.113 | −0.136 | −0.128 | −0.002 | **−0.286**\*\* | −0.065 |
| Height | −0.024 | 0.112 | 0.067 | −0.081 | −0.009 | −0.004 |
| Body shape | 0.051 | 0.103 | 0.088 | 0.065 | **0.323**\*\* | −0.086 |

Relative weight is participant weight controlled for height (i.e., equivalent to body mass index if the relationship is isometric). Bold values indicate statistical significance. Asterisks indicate FDR rate at ***$P<0.001$, **$P<0.01$, *$P<0.05$ (see Methods).

**Table 3. Results from general linear mixed models with parameter estimates and chi-squared ($\chi^2$) test statistics (all $df=1$), including response time (only using values ≤20 s) as the dependent variable and the three standardized male traits as fixed predictors for: (a) female rating of attractiveness; (b) male rating of attractiveness; and (c) male rating of fighting ability for in-person and online data.**

| In-person | | | | Online | | | |
|---|---|---|---|---|---|---|---|
| Traits | Estimate | $\chi^2_1$ | $P (\chi^2)$ | Traits | Estimate | $\chi^2_1$ | $P (\chi^2)$ |
| **(a) Female rating of male attractiveness** | | | | | | | |
| (Intercept) | 7.869 | | | (Intercept) | 8.525 | | |
| Penis size | 0.027 | 15.431 | **<0.0001** | Penis size | 0.034 | 30.217 | **<0.00001** |
| Height | 0.024 | 12.452 | **0.0004** | Height | 0.015 | 5.584 | **0.018** |
| Body shape | 0.11 | 257.851 | **<0.0001** | Body shape | 0.047 | 59.973 | **<0.00001** |
| **(b) Male rating of a rival's attractiveness** | | | | | | | |
| (Intercept) | 7.832 | | | (Intercept) | 8.057 | | |
| Penis size | 0.049 | 36.048 | **<0.00001** | Penis size | 0.025 | 12.568 | **0.0004** |
| Height | 0.031 | 14.361 | **0.0002** | Height | 0.009 | 1.561 | 0.212 |
| Body shape | 0.084 | 107.622 | **<0.00001** | Body shape | 0.042 | 35.908 | **<0.00001** |
| **(c) Male rating of a rival's fighting ability** | | | | | | | |
| (Intercept) | 7.558 | | | (Intercept) | 8.142 | | |
| Penis size | 0.001 | 0.006 | 0.939 | Penis size | 0.024 | 12.859 | **0.0003** |
| Height | 0.048 | 38.135 | **<0.00001** | Height | 0.009 | 1.63 | 0.202 |
| Body shape | 0.025 | 10.425 | **0.001** | Body shape | 0.035 | 27.319 | **<0.00001** |

Bold values indicate statistical significance with original *P* values after accounting for a false discovery rate for multiple comparisons at the level of analysis.

## Female rating of male attractiveness

Females rated a more V-shaped body, being taller and a larger penis as more sexually attractive (**Table 1a**). There was highly significant positive linear selection on penis size, height, and the shoulder-to-hip ratio based on female rating of male attractiveness for both the in-person and online surveys. Linear selection was strongest on male body shape, with weaker but still significant selection on height and penis size. There were, however, diminishing benefits of increased penis size, height, or a greater shoulder-to-hip ratio in elevating attractiveness. This decline is evident from significant quadratic selection gradients for these traits (except for height in the online survey; see **Table 1a** and **Figs 2A**, **2B**, **3A**, **3B**, **4A and 4B**). Non-linear selection was strongest on body shape, and almost identical for height and penis size (**Table 1a**).

Body shape, height and penis size interacted to affect how females rated male attractiveness (**Table 1a**). There were significant correlational selection gradients for all trait combinations, except for that between penis size and height for the online survey. In the in-person survey, after controlling for body shape, greater penis size elevated attractiveness more strongly for taller men (**Fig 5A**). Likewise, for both the in-person and online surveys, after controlling for height, there was a significant increase in attractiveness with penis size for men with a greater shoulder-to-hip ratio (**Fig 5B**). And for both the in-person and online surveys, after controlling for penis size, a greater shoulder-to-hip ratio increased attractiveness more for taller men (**Fig 5C**).

## Male rating of a rival's attractiveness

Males rated a more V-shaped body, being taller and having a larger penis as more sexually attractive to females (based on how jealous they would be if the figure talked to their female partner; **Table 1b**). As with female rating of male attractiveness, there was highly significant positive linear selection on penis size, height, and the shoulder-to-hip ratio for both

PLOS Biology

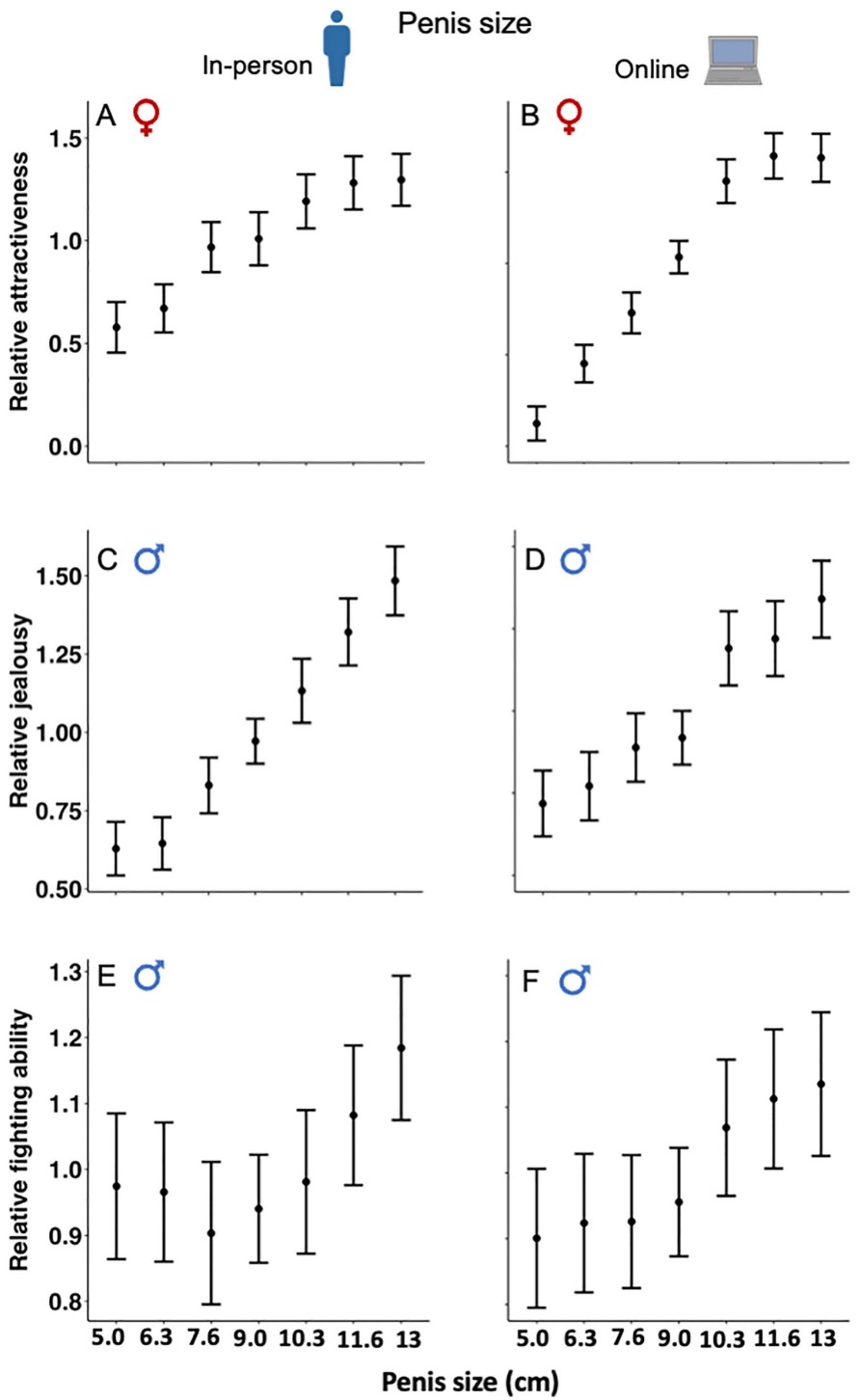

**Fig 2. Relationship between participant rating and penis size for in-person and online surveys: (A, B) female rating of male attractiveness; (C, D) male rating of rival's attractiveness; and (E, F) male rating of rival's fighting ability.** The mean values represent standardized participant rating, and the upper and lower values are 95% confidence intervals. Note that the range of the Y-axis differs for the three types of surveys. The data underlying this Figure can be found in S1 Data.

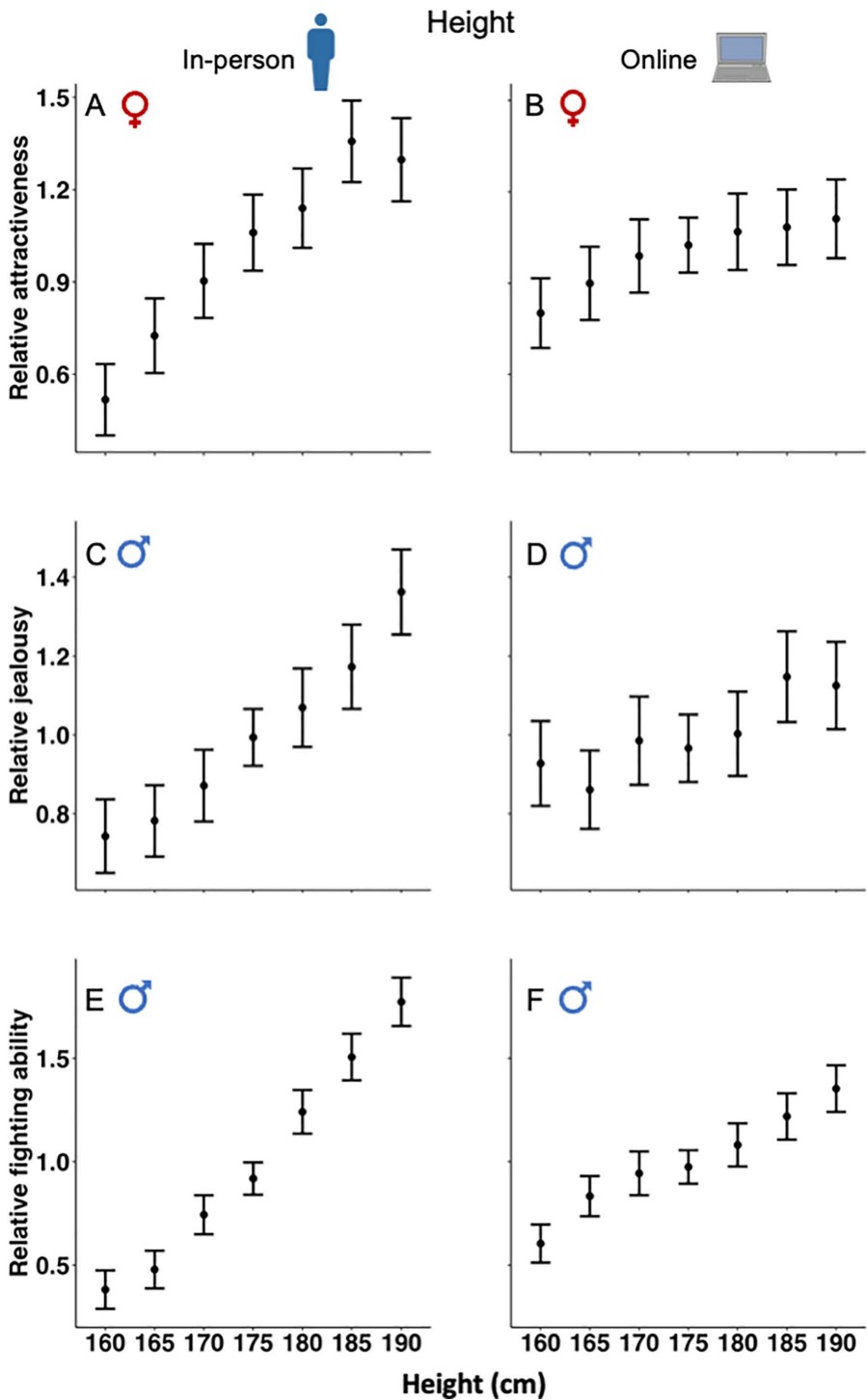

**Fig 3. Relationship between participant rating and height for in-person and online surveys: (A, B) female rating of male attractiveness; (C, D) male rating of rival's attractiveness; and (E, F) male rating of rival's fighting ability.** The mean values represent standardized participant rating, and the upper and lower values are 95% confidence intervals. Note that the range of the Y-axis differs for the three types of surveys. The data underlying this Figure can be found in S1 Data.

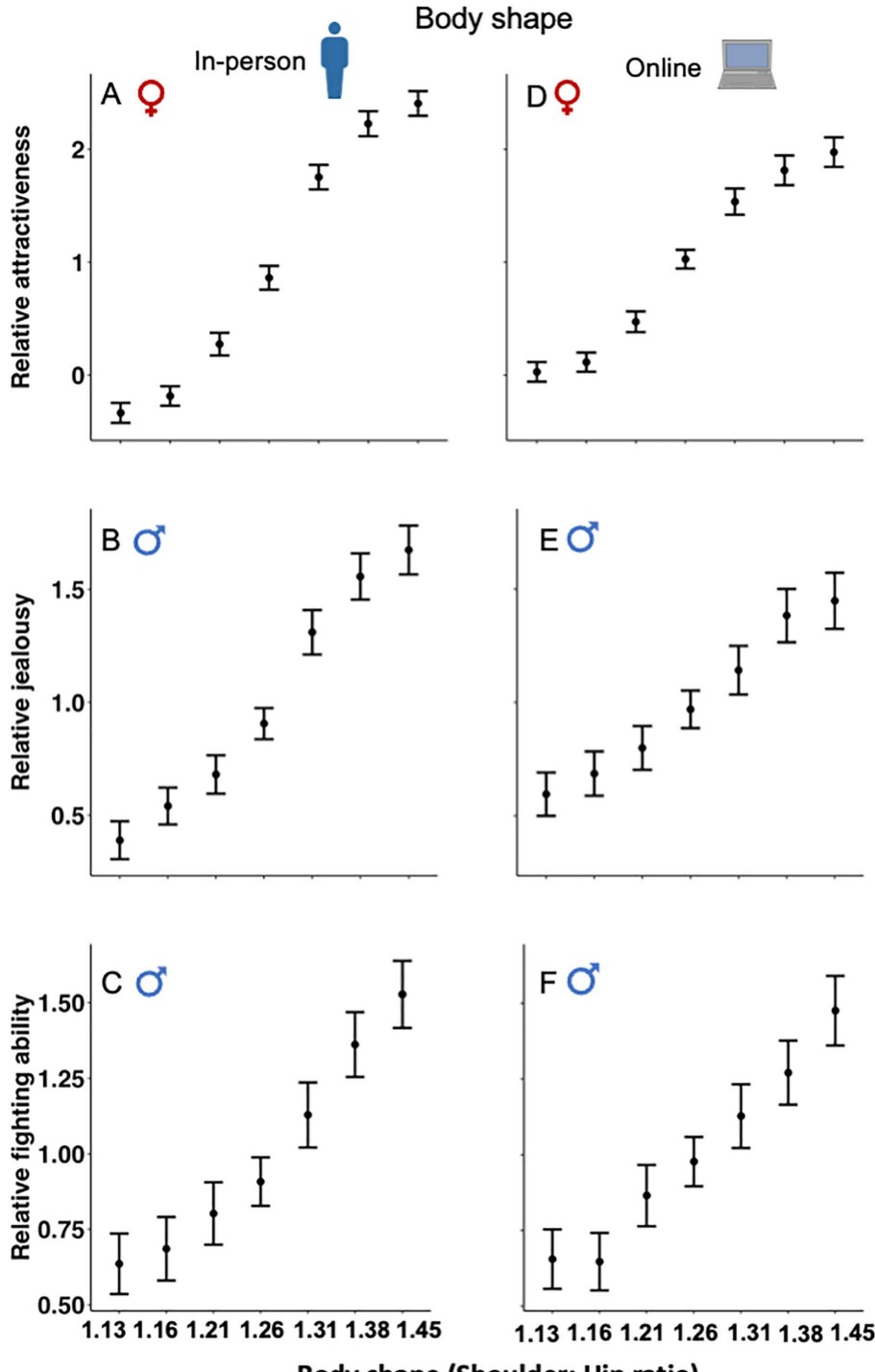

**Fig 4. Relationship between participant rating and body shape for in-person and online surveys: (A, B) female rating of male attractiveness; (C, D) male rating of rival's attractiveness; and (E, F) male rating of rival's fighting ability.** The mean values represent standardized participant rating, and the upper and lower values are 95% confidence intervals. Note that the range of the Y-axis differs for the three types of surveys. The data underlying this Figure can be found in S1 Data.

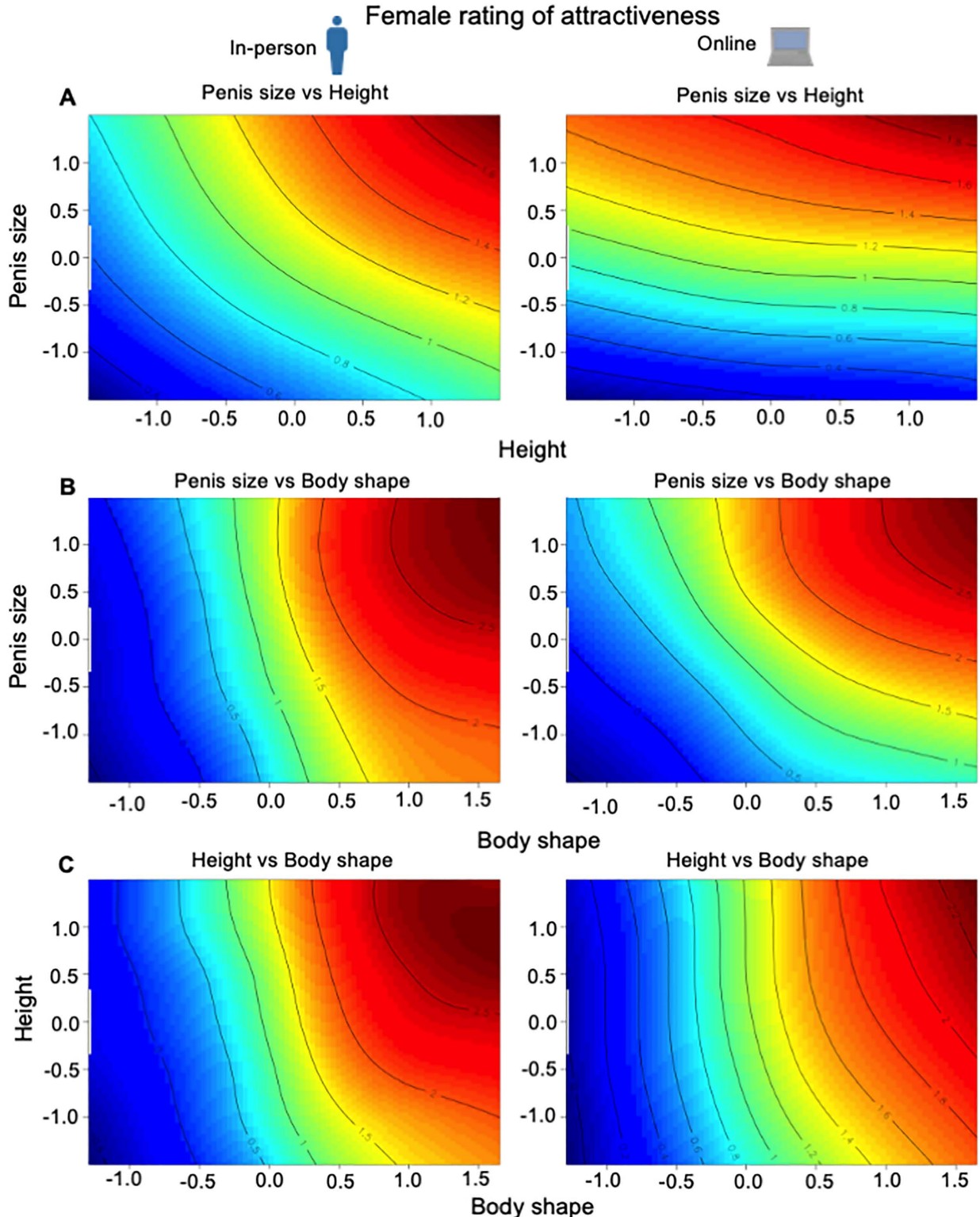

**Fig 5. Contour map of the fitness surface (red: greater attractiveness) for female rating of attractiveness for in-person and online surveys for: (A) penis size and height (body shape controlled); (B) penis size and body shape (height controlled); and (C) height and body shape (penis size controlled).** Contour line of 1 = mean attractiveness. The data underlying this Figure can be found in S1 Data.

the in-person and online surveys. Again, linear selection was strongest on body shape, with weaker but still significant selection on height and penis size. Unlike the case for female ratings, however, there was no evidence for diminishing effects of increased penis size, being taller or a greater shoulder-to-hip ratio on male assessment of a rival's attractiveness to females. This is evident from non-significant quadratic selection gradients (**Table 1b** and **Figs 2C**, **2D**, **3C**, **3D**,**4C** and **4D**).

As with female rating of male attractiveness, body shape, height and penis size interacted to affect male rating of another male's attractiveness. There were significant correlational selection gradients involving body shape for both the in-person and online surveys (**Table 1b**). Controlling for height, there was a significant increase in attractiveness with penis size for men with a greater shoulder-to-hip ratio (**Fig 6B**). Controlling for penis size, a greater shoulder-to-hip ratio increased attractiveness more strongly for taller men (**Fig 6C**). Unlike female ratings, however, the interaction between penis size and height was not significant (**Fig 6A**).

### Male rating of a rival's fighting ability

Males rated a more V-shaped body, being taller, and a larger penis as indicative of greater fighting ability (based on how threatened the participant would be if the male figure initiated a fight; **Table 1c**). Again, there was highly significant positive linear selection on penis size, height, and the shoulder-to-hip ratio for both the in-person and online surveys. Selection was strongest on height and weakest on penis size for the in-person survey; and selection was strongest on the shoulder-to-hip ratio and weakest on penis size for the online survey. There was a diminishing benefit to increased height for the in-person survey, but there was otherwise no evidence for non-linear selection as the other five quadratic selection gradients were non-significant (**Table 1c** and **Figs 2E**, **2F**, **3E**, **3F**, **4E**, **and** **4F**).

Body shape, height, and penis size interacted to affect male rating of fighting ability. Correlational selection was, however, weaker than that for either male or female assessment of attractiveness, and it did not involve interactions with penis size (**Table 1c** and **Fig 7A** and **7B**). The only significant correlational selection gradient was for height and body shape. In both the in-person and online surveys, after controlling for penis size, a greater shoulder-to-hip ratio increased male rating of fighting ability more strongly for taller men (**Fig 7C**).

### Participant traits analysis

The correlations between participant traits and the linear selection coefficients for the three focal male traits are presented in **Table 2**.

**Female rating of male attractiveness.** Male height more strongly influenced ratings of attractiveness among taller females (in person: $r=0.321$, $P=0.0008$; online: $r=0.269$, $P=0.011$), but there was no such relationship for penis size or body shape. Penis size also more strongly influenced attractiveness ratings for heavier females completing the in-person survey ($r=0.329$, $P=0.0006$), but not for the online survey ($r=−0.077$, $P=0.477$). A female's body mass index did not influence the strength with which height or body shape affected how she rated male attractiveness. Female age did not influence how strongly penis size, height, or body shape affected her rating of male attractiveness (all: $P>0.02$ after correcting for FDR).

**Male rating of a rival's attractiveness.** A rival's height more strongly influenced ratings of attractiveness for taller men completing the in-person survey ($r=0.402$, $P=0.00003$), but not for the online survey ($r=0.101$, $P=0.348$); and there was no such relationship for penis size or body shape. Rival's penis size also more strongly influenced attractiveness ratings for older males completing the in-person survey ($r=0.322$, $P=0.001$), but not the online survey ($r=−0.093$, $P=0.386$). In contrast, older males were significantly less influenced by a rival's body shape when rating his attractiveness for both the in-person ($r=−0.297$, $P=0.003$) and online surveys ($r=−0.301$, $P=0.004$). Older males did not have a significant tendency to be more strongly influenced by the height of a rival when rating his attractiveness for either the in-person or online surveys.

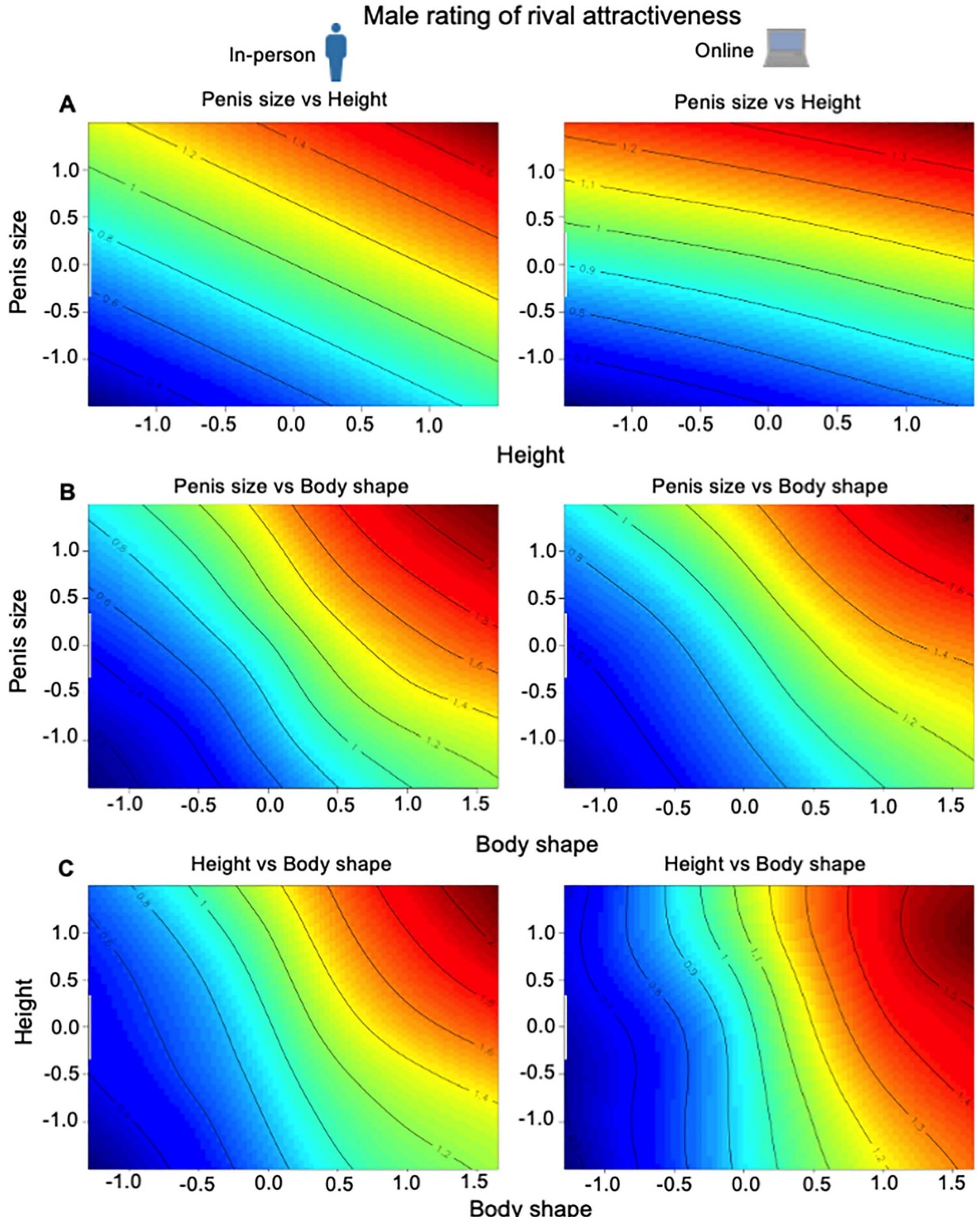

**Fig 6. Contour map of the fitness surface (red: greater attractiveness) for male rating of rival's attractiveness for in-person and online surveys for: (A) penis size and height (body shape controlled); (B) penis size and body shape (height controlled); and (C) height and body shape (penis size controlled).** Contour line of 1 = mean attractiveness. The data underlying this Figure can be found in S1 Data.

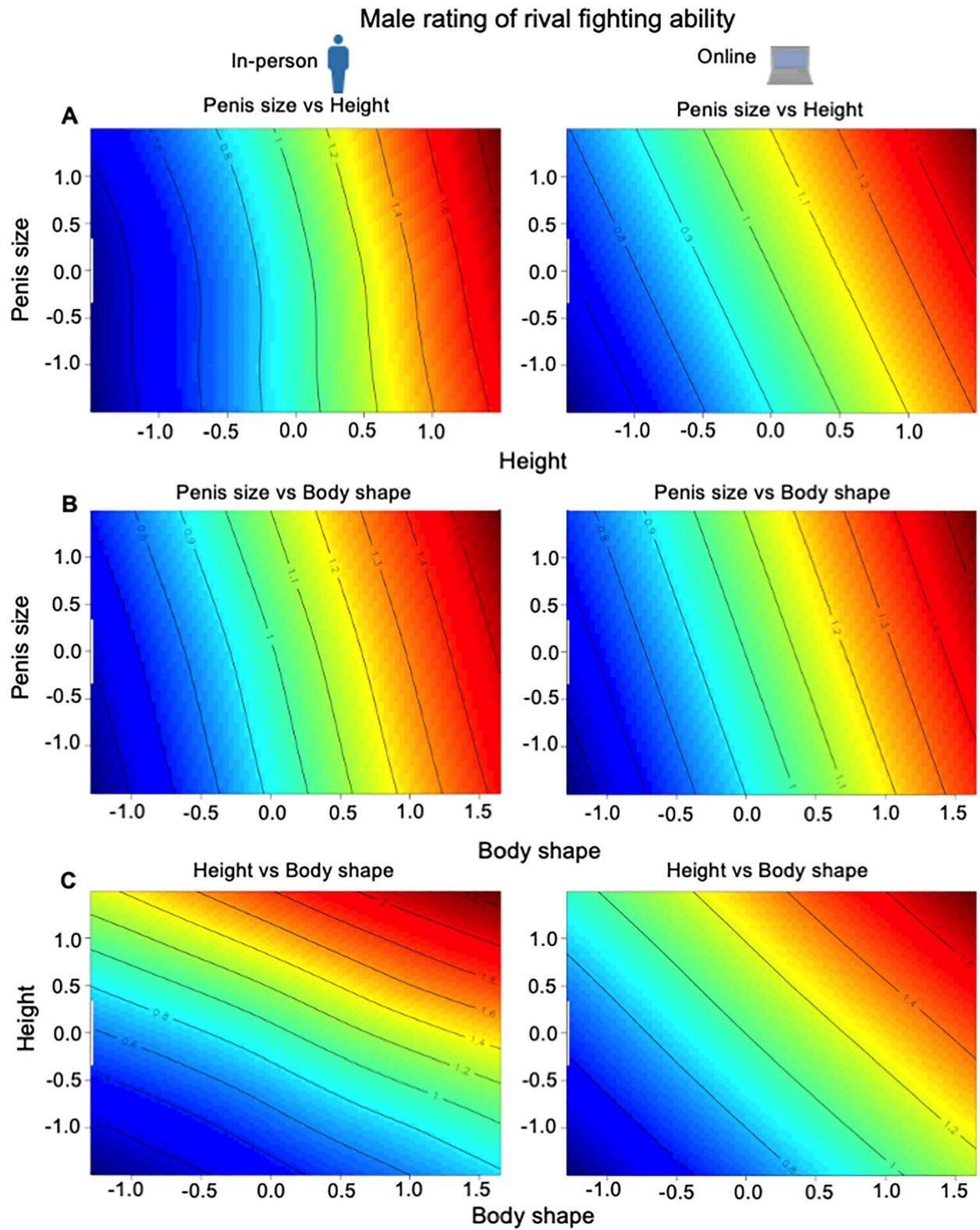

**Fig 7. Contour map of the fitness surface (red: greater fighting ability) for male rating of rival's fighting ability for in-person and online surveys for: (A) penis size and height (body shape controlled); (B) penis size and body shape (height controlled); and (C) height and body shape (penis size controlled).** Contour line of 1 = mean fighting ability. The data underlying this Figure can be found in S1 Data.

**Male rating of a rival's fighting ability.**  For the in-person survey, neither a male's age, height or body mass index influenced how strongly a rival's penis size, height, or body shape affected his rating of fighting ability (all $P > 0.02$ after correcting for FDR). For the online survey, taller males were significantly less influenced by the rival's penis size ($r = -0.286$, $P = 0.005$), and significantly more influenced by the rival's body shape ($r = 0.323$, $P = 0.002$), while the rival's height had no effect.

## Response time analysis

The relationships are summarized in **Table 3**.

*Female rating of male attractiveness.*  The response time was significantly faster for figures with a smaller penis, shorter height, and a less V-shaped body for both in-person and online surveys (i.e., faster responses to less attractive figures that were given lower scores).

**Male rating of a rival's attractiveness.**   The response time was significantly faster for figures with a smaller penis, shorter height, and a less V-shaped body for both in-person and online surveys; the only exception being that height did not significantly affect response time in the online survey ($P = 0.212$).

**Male rating of a rival's fighting ability.**  The response time was significantly faster for figures with a smaller penis for the online survey, but not for the in-person survey ($P = 0.939$). The response time was significantly faster for figures with a less V-shaped body for both the online and in-person surveys. The response time was significantly faster for shorter figures for the in-person but not for the online survey ($P = 0.202$).

## Discussion

There was a consistent pattern as to how penis size, height, and body shape affected female assessment of male attractiveness, male assessment of a rival's attractiveness (sexual competitiveness), and male evaluation of a rival's fighting ability. For all three focal traits, being taller, having a more V-shaped body, and having a larger penis increased perceptions that a male was more attractive and had greater fighting ability. Consistent with the ranking score given to figures, both male and female participants also took significantly less time to assign a (lower) score to figures with a smaller penis, shorter height, and a less V-shaped body. The main effects of the three focal traits were similar for both the in-person surveys rating life-sized figures, and the online surveys rating smaller figures. There were, however, some differences between in-person and online surveys, mostly characterized by significant effects detected in the in-person survey but not in the online survey (though in three analyses, the reverse was true). Both male and female participants generated linear selection on all three focal traits, but there was a clear trend for females to generate stronger non-linear and correlational selection on male morphology. For females, 10 of 12 estimates were significant when they rated male attractiveness, while for males only two of 12 estimates were significant when they rated a rival's attractiveness, and only three of 12 estimates were significant when they rated a rival's fighting ability.

Crucially, we always detected significant linear selection on penis size despite differences between the two types of surveys in the strength of non-linear selection on traits. A larger penis increased male attractiveness to females and was also used as a signal of fighting ability by males. By re-analyzing the findings of [35] alongside our new data on male assessment of rival figures and the use of both in-person and online surveys of participants of both sexes, our study provides evidence that pre-copulatory sexual selection due to both female choice and male-male competition might have contributed to the enlarged size of the human penis when compared to that of other primates.

### The effect of survey type

We assume that in-person surveys are more likely than online surveys to reflect actual selection because participants are assessing life-sized figures, which is more biologically realistic. The observed differences in the magnitude of effect sizes between the two types of surveys might reflect greater difficulties in accurately detecting variation in size when viewing the

small figures viewed during online surveys compared to the life-sized figures viewed during in-person surveys [30,51]. For example, life-sized figures provide participants with additional referential cues, such as having to look upward to view the face if a figure is taller than the participant.

Selection on height was more strongly affected by the survey type than selection on penis size or body shape. Linear selection on height was 2- to 3-fold stronger for in-person than online surveys for female assessment of attractiveness, male perception of attractiveness, and male estimation of fighting ability (**Table 1**). Likewise, eight of nine quadratic or correlational selection gradients for height were stronger for the in-person than online surveys. In contrast, there was no consistent bias in the magnitude of selection coefficients due to survey type for estimates of non-linear selection on penis size or body shape. The absence of an effect of survey type on estimates of non-linear selection on penis size, despite it being a size-based trait, might reflect the fact that body size is used as reference against which to assess penis size.

In general, we suggest that it is advisable to use in-person surveys and life-sized stimuli to measure sexual selection on morphological traits. Our findings indicate that online surveys might underestimate, or even fail to detect, sexual selection. For example, the online surveys failed to identify that greater male height has diminishing effects on female assessment of attractiveness and male assessment of fighting ability. An obvious reason for the reported differences between the survey types is that the size of traits is more difficult to assess when they are presented as small images. A less obvious reason is that participants in in-person surveys are tested in a neutral setting and might also be more motivated to complete the task (i.e., they had to travel to the survey venue) than participants completing the survey at home who have paid a smaller time cost, and are in a setting that is likely to be more distracting [50]. In sum, our results suggest that online surveys can limit the ability of researchers to detect diminishing returns of trait exaggeration; online surveys can also reduce the ability of researchers to detect how traits interact to affect assessment of mates or rivals. Unfortunately, most studies of sexual selection in humans still present small rather than life-sized images [ 33,36,45].

### Sex differences in evaluating attractiveness

Male and female evaluations were broadly similar when assessing the attractiveness of males. The main difference was that the focal traits had a stronger effect on female scores. For example, five of six estimates of linear selection were larger for female than male evaluations (**Table 1**); and of the 10 of 12 estimates of non-linear selection that were significant based on female evaluation, only four were significant based on male evaluation. Intriguingly, there was a difference in the shape of the selection surface generated by females versus males. We found strong evidence for directional selection with diminishing returns for greater expression of traits when women were assessing male attractiveness. While taller, more V-shaped men with larger penises were consistently preferred, the fitness benefits of further exaggeration began to plateau at the upper end of the trait values. However, there was no evidence for diminishing returns on penis size, height and body shape based on male assessment of attractiveness. The selection could therefore be described as open-ended, directional selection, where the perceived attractiveness of a rival continued to increase linearly with the exaggeration of his traits. This sex-specific difference in evaluation suggests that males tend to overestimate the extent to which these traits affect female mate choice, thereby inflating their perception of the level of competition posed by more attractive rivals.

In general, our results suggest that men can identify which traits women prefer, but they do so with less accuracy. This supports the claim that men can assess the competitive mating market by gauging their relative value as mates [52,53]. This ability should then cause men to adjust their mating strategies accordingly, as occurs in other animals where, for example, less competitive individuals often pursue alternative mating tactics to circumvent female mate choice [54,55]. One reason why males might overestimate the importance of the focal traits is that they also use these traits to evaluate males as opponents during aggressive encounters (see below) and therefore conflate two aspects of male competitiveness of mating rivals [56]. This tendency could also reflect an adaptive bias driven by the asymmetric costs of misjudging what women find attractive (i.e., error management theory; [57]). Men may overestimate the extent to which the focal traits

we manipulated affect female mate choice because the cost of neglecting these traits may reduce mating success more than the cost of over-investing in them.

## Penis size and assessment of fighting ability

We believe that the most important finding of our study is that penis size influences how males assess a rival's fighting ability. Males were more likely to feel threatened if they had to imagine being challenged by a rival with a large penis. We infer that this translates into males being less likely to initiate an aggressive interaction with a rival with a large penis. We therefore provide the first robust experimental evidence that penis size is under precopulatory sexual selection in humans because it can be used as a signal of male fighting ability. One possible explanation why a larger penis might be an honest signal of fighting ability is that testosterone influences penile development at puberty, and higher levels of testosterone in adult males are associated with increased muscle mass, greater aggression, and higher competitive ability [42,58,59]. A second, non-mutually exclusive explanation is that flaccid penis length functions as an indicator of a male's physiological state. Flaccid penis length can shorten in response to stress or anxiety as adrenaline redirects blood flow away from the genitals, a response that may function to reduce injury during 'fight or flight' situations [60,61]. Therefore, participants may interpret a flaccid penis that is longer as a signal of rival confidence, lower stress, or the relative absence of perceived threat.

It should, however, be noted that selection on penis size, because it elevates perceived fighting ability, was much weaker than selection on height or body shape (3- to 8-fold less; **Table 1**). Assessment of the strength of a rival is therefore mainly determined by his height and body shape. Likewise, the effect of penis size on male-perceived fighting ability was much weaker (4- to 7-fold less; **Table 1**) than its effect on attractiveness to females. This finding is intriguing as opposite patterns have been reported for other male secondary sexual traits, such as beards [62] and deep voices [63], which more strongly influence male perceptions of dominance and fighting ability than attractiveness to females. This contrast suggests that while many sexually dimorphic male traits function primarily as "badges of status" for male-male contests in humans [64–66], penis size might function primarily as a "sexual ornament" for attracting mates. This aligns with our findings that traits like height and body shape explains evaluation of a rival's fighting ability, whereas penis size seems to have a stronger effect as a display trait that affects female assessment of attractiveness. Indeed, while body shape was the strongest driver of male attractiveness to females, penis size and height had very similar effects on male attractiveness for the in-person survey, and penis size actually had a far stronger effect than height for the online survey. While our study confirms the role of penis size in assessment of a rival's fighting ability, the main source of sexual selection for enlargement seems to be female choice.

## Participants' own traits

Some of the participant's own traits significantly influenced how they rated male figures. Taller females were significantly more strongly influenced by male height when assessing attractiveness. This was the case in both the in-person and online surveys. This finding is consistent with females preferring a partner who is taller than themselves, which contributes to assortative mating by height in human populations [67,68]. Females who were heavier than normal for their height also had a significantly stronger preference for a larger penis, but this relationship was only apparent for the in-person survey.

For male participants, the only relationship observed for both the in-person and online surveys is that a more V-shaped body of a rival significantly more strongly influenced younger than older males when evaluating a rival's attractiveness. This finding is in line with those of [63], who found low voice pitch (another dominance-related trait) more strongly influenced perceptions of fighting ability in younger than older men. One explanation for such results is that younger males are more likely to engage in physical fights and therefore assess traits associated with strength and dominance, while older males may rely more on social status or coalitions to prevent fights occurring, and instead assess traits linked to reproductive success when they evaluate rivals [69]. For the in-person surveys, penis size significantly and more strongly

influenced older than younger males when assessing the attractiveness of a rival. This suggests that perceptions of sexual competition might increase with age, perhaps driven by experience gained in how females evaluate male attractiveness. There was, however, no equivalent relationship for the online survey. Finally, taller males were significantly more attentive to height when evaluating a rival's attractiveness. This result adds to the existing literature exploring how a man's own dominance affects his perception of rivals. The results of studies have been mixed. For example, some studies have found that shorter or less dominant men are more sensitive to dominance cues in rivals [70], whereas others found no such detectable relationship (e.g., [71]). One explanation is that shorter men do not need to distinguish between slightly taller and much taller than average males because they are both more attractive than a short man. Again, however, this pattern was apparent for our in-person survey but not for our online survey.

**Human penis size and sexual selection**

Our study provides insights into the potential role of sexual selection in the evolution of penis size in humans. It is, however, important to recognize that human mate preferences are strongly influenced by cultural norms and the environmental setting [72]. Ideals about masculinity and attractiveness vary widely across cultures and over time within different cultures. Some societies place great emphasis on height and body shape, while others prioritize facial features or social status [73]. It is worth noting that while the data for the in-person female surveys was mostly from white females [35], the other five surveys we ran involved participants from a wide range of ethnicities. It should further be noted that selection analysis at the population-level (i.e., using mean scores per figure) closely resembled those seen when calculating individual selection gradients for each participant. This is particularly apparent for linear selection. This suggests that any variation among participants due to ethnicity, cultural upbringing and so on, is minor. Most participants show the same pattern of evaluating taller, more V-shaped males with a larger penis as more attractive or better fighters. It is important to note that our computer-generated figures had flaccid penises. Flaccid penis length is known to be positively correlated with erect length [60]. While a flaccid penis is a more common state in humans and probably a more relevant signal available to assess a rival's fighting ability, male-male displays in other primates and female choice in humans can involve erect penises [1,2]. It is plausible, therefore, that using figures with an erect penis might reveal an even stronger effect of penis size on sexual attractiveness and assessment of fighting ability. Future studies using erect penises remain an avenue for future research to test this claim. Of course, the degree to which penis size is emphasized varies with media portrayals and societal narratives. Future studies could therefore include cross-cultural comparisons to disentangle the relative contribution of cultural norms to selection on male traits due to mate choice or fighting assessment.

In sum, we provide robust experimental evidence that sexual selection can act on multiple male traits that combine to determine male attractiveness and the perception of attractiveness and fighting ability. These traits include penis size, in association with height and body shape. Controlling for variation in other traits, both female mate choice and male assessments of rivals strongly favor larger penis size. The question remains open, however, as to whether this potential for selection affected the evolution of human penis size, as the opportunity for female choice is unresolved in our ancestral past. In patrilineal and agropastoral societies, marital decisions are usually determined by kin. In traditional foraging societies—commonly used as proxies for ancestral human social structures—parents generally exert significant control over their daughters' mating decisions [74]. In matrilineal and matrilocal systems, however, female autonomy is more pronounced [75]. Likewise, across many contexts, women often exercised mate choice even under constraint—through selective compliance, extra-pair relationships, or re-partnering after marital dissolution [76]. The existence of the strong female preference for greater penis size that we observed in our study suggests that these preferences are likely to have been present in our ancestors, and therefore will have had some role in sexual selection, causing the evolution of the enlarged human penis. It is also worth noting that penis size and shape may have been selected for because of their effects on sexual stimulation, specifically by increasing the likelihood of female orgasm, which has, in turn, been linked to improved sperm retention, facilitating sperm activation, and encouraging additional copulations [6,77,78]. It therefore

seems likely that both pre- and post-copulatory sexual selection have contributed to the evolution of human penis size. Given that little is known about the link between penis size and the outcome of male-male contests for mates, this link is worthy of further exploration as a potential driver of the evolution of penis size in our ancestors [79].

## Supporting information

**S1 Text. Supplementary tables.** Table A in S1 Text. Final sample size per survey. For each survey, data were collected from >100 participants, but some participants were excluded (see Materials and methods). Table B in S1 Text. Outputs from likelihood ratio tests assessing differences in the selection matrix due to mode of delivery, participant sex, and assessment factor (attractiveness vs. fighting ability). Bold values indicate statistical significance after FDR correction. Table C in S1 Text. Linear (β) and quadratic/correlational (γ) selection gradients based on means of gradients generated for each participant for in-person (A) and online (B) surveys. Asterisks indicate significance after FDR correction. Table D in S1 Text. General linear mixed model results for all response time as the dependent variable and the three standardized male traits as fixed covariates for in-person and online surveys. Bold values indicate statistical significance after FDR correction.
(DOCX)

**S1 Appendix. Analyses of all paid online data.**
(DOCX)

**S1 Data. Data and R code used for all analyses in this study.**
(ZIP)

## Author contributions

**Conceptualization:** Chloe Tan, Rebecca Bathgate, Michael D. Jennions.

**Data curation:** Upama Aich, Chloe Tan, Robert C. S. Capp, Jacob C. Kuek.

**Formal analysis:** Upama Aich.

**Funding acquisition:** Upama Aich, Bob B. M. Wong.

**Investigation:** Upama Aich, Chloe Tan, Rebecca Bathgate, Khandis R. Blake, Michael D. Jennions.

**Methodology:** Chloe Tan, Brian S. Mautz.

**Project administration:** Rebecca Bathgate, Michael D. Jennions.

**Resources:** Upama Aich, Khandis R. Blake, Brian S. Mautz.

**Supervision:** Michael D. Jennions.

**Visualization:** Upama Aich, Brian S. Mautz.

**Writing – original draft:** Upama Aich.

**Writing – review & editing:** Upama Aich, Chloe Tan, Rebecca Bathgate, Khandis R. Blake, Robert C. S. Capp, Jacob C. Kuek, Bob B. M. Wong, Brian S. Mautz, Michael D. Jennions.

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
