## [Editor Report · Decision Letter 0]

4 Sep 2025

Dear Dr Aich,

Thank you for submitting your manuscript entitled "Effect of penis size, height and body shape on sexual attractiveness, perceived attractiveness and assessment of fighting ability in human" for consideration as a Research Article by PLOS Biology.

Your manuscript has now been evaluated by the PLOS Biology editorial staff, as well as by an academic editor with relevant expertise, and I'm writing to let you know that we would like to send your submission out for external peer review.

Once your full submission is complete, your paper will undergo a series of checks in preparation for peer review. After your manuscript has passed the checks it will be sent out for review. To provide the metadata for your submission, please Login to Editorial Manager (https://www.editorialmanager.com/pbiology) within two working days, i.e. by Sep 08 2025 11:59PM.

Kind regards,

Roli Roberts

Roland Roberts, PhD

Senior Editor

PLOS Biology

rroberts@plos.org

---

## [Decision Letter · Decision Letter 1]

31 Oct 2025

Dear Dr Aich,

Thank you for your patience while your manuscript "Effect of penis size, height and body shape on sexual attractiveness, perceived attractiveness and assessment of fighting ability in human" went through peer-review at PLOS Biology. Your manuscript has now been evaluated by the PLOS Biology editors, an Academic Editor with relevant expertise, and by three independent reviewers.

You'll see that reviewer #1 thinks that you should clarify the relationship to the 2013 PNAS paper, and finds the results convincing but unsurprising. Reviewer #2 is also positive, but again wants clarity around the fact that part of the study replicates the PNAS study, asks about erect vs flaccid penises, makes a suggestion about showing paired stimuli in the online study (I assume he means in the future, so solely to discuss), and has a number of minor requests. Reviewer #3 is again positive, but has a number of substantial suggestions around the framing and interpretation.

In light of the reviews, which you will find at the end of this email, we are pleased to offer you the opportunity to address the comments from the reviewers in a revision that we anticipate should not take you very long. We will then assess your revised manuscript and your response to the reviewers' comments with our Academic Editor aiming to avoid further rounds of peer-review, although we might need to consult with the reviewers, depending on the nature of the revisions.

IMPORTANT - please attend to the following:

a) Please can you change the Title to include an active verb? We suggest: "The penis size, height and body shape of human males influence sexual attractiveness, perceived attractiveness and assessment of fighting ability"

b) Please attend to all of the requests from the reviewers.

c) Please ensure that you comply with our Data Policy; specifically, we need you to supply the numerical values underlying Figs 2ABCDEF, 3ABCDEF, 4ABCDEF, 5ABC, 6ABC, 7ABC, either as a supplementary data file or as a permanent DOI’d deposition. My understanding is that the data and code provided in S1 Data are sufficient to recreate the Figures; please can you confirm whether this is the case?

d) Please cite the location of the data clearly in all relevant main and supplementary Figure legends, e.g. “The data underlying this Figure can be found in S1 Data” or “The data underlying this Figure can be found in https://zenodo.org/records/XXXXXXXX

**IMPORTANT - SUBMITTING YOUR REVISION**

*Resubmission Checklist*

*Published Peer Review*

*PLOS Data Policy*

Sincerely,

Roli Roberts

Roland Roberts, PhD

Senior Editor

PLOS Biology

rroberts@plos.org

REVIEWERS' COMMENTS:

Reviewer #1:

In this manuscript the authors use over computer-generated images displayed life size in person or online to test how male genital size, height and body size influences assessment of a male's attractiveness by females, and males' perception of fighting ability and attractiveness to females. The design allows the authors to separately determine the affect of the three traits independently aa well as interactions between them. The main new result is that males use male genital size to assess male fighting ability and attractiveness to females. This suggests that male-male competition may have played a role in the evolution of the unusually large genitalia of human males.

I do not have any substantial criticism of this paper. The analyses are competent and thorough and the results are clear. I do think the authors could make it a little more clear why the female data is included here given that the results are those published some years ago in PNAS. They do not this, but I think it could be made much more clear in the discussion, including a statement of what it has added to the current analyses.

I have marked "accept" in my recommendation. However, I have to note that I was unsurprised by the results and I do not think many in the field will be surprised. My view of this is that height, broad shoulders and genital size are all traits that distinguish males. That females prefer "male" traits and the same traits indicate competition in males is not particularly surprising to me, given our broader knowledge of sexual selection in animals.

Though the results are unlikely, in my opinion, to surprise those in the field, the study is strong in methods and analyses, and it will surely attract substantial attention in the general science press.

Reviewer #2:

[identifies himself as Bogusław Pawłowski]

This is very interesting paper with quite rigid methods used by the authors. The results are very interesting in a few aspects. Firstly, when controlling for other studied traits they found that a man's height, body shape and penis size are related to perceived attractiveness and assessment of fighting men ability by other males. It supports the hypothesis on the sexual selection for these three traits. Secondly, they found that the way the study is conducted (online vs in person) may affect some results. Might be it is rather a matter of showing stimuli in life sizes (in the in-person study), which method looks as more relevant for such studies (what the authors also underline). Thirdly, as could be expected it is body height and body shape that are more important both for attractiveness and the assessment of fighting ability, than penis size itself. Fourthly, they showed sex differences in the sexual selection pressure which appeared to be a bit stronger in the case of male intrasexual selection ("…there was no evidence for diminishing effects (as it was in the case of women assessing attractiveness) of increased penis size, being taller or a greater shoulder-to-hip ratio on male assessment of other males' attractiveness to females"). Fifth, they showed that assessments might depend on a subject's own body morphology (or in one case a male subject age). What is also interesting, is that response time was related to the perceived stimuli traits. Might be the author could then mention in the Discussion that the perception in the studied aspects is "better tuned" to exclude some traits (this aligns better with the "bad genes hypothesis" than with "good genes hypothesis") than to choose the best one from the "good ones".

As far as attractiveness assessments by women is concerned, it seems to by rather replication of the study on exactly the same stimuli published in 2013 (https://doi.org/10.1073/pnas.1219361110). I think it should be then clearly written in the paper that this part of the present study in fact replicates the previous study. The original part of the present study is related to both males' assessments (the level of sexual competition "jealous" and fighting ability).

The stimuli used by the authors have flaccid penises and since it has been already shown that flaccid penis length is positively related to the penis size in erection, the authors should refer to the adequate papers (some references are needed). It is because that it is the erect penis (as in some other primates) that can be used to deter e.g. males from other group (and not the flaccid one). The question is then if the stimuli with the erect penises could reveal stronger effect than the stimuli with the flaccid penises.

The discussion of the results and the difference between online and in-person results is convincing. To make height more relevant stimulus in online study, I would suggest rather showing pairs (having the chance of comparison in one picture) in which e.g. the shorter silhouette would be the imaginable size of a subject which is supposed to assess a rival attractiveness or fighting ability. This is a method used when studying preference of sexual dimorphism in height between partners, and in the case of the present study it can be converted e.g. to two men silhouettes. The result that taller women pay more attention for a man height, confirms previous studies (taller male norm in WEIRD countries).

Minor points:

There is 73 = 343 and should be 73 = 343

Does in online study the subjects also saw each time only one picture (gif) at a time as in the in-person study - it should be clarified.

On p. 15 it is "males rating male attractiveness" - which seems not be true according to the previous information ("competitiveness")

The authors write: "We used the unpaid surveys to reduce the risk that some participants did not pay attention to the figures and only completed the survey for financial gain. T". So, what is the sense to collect the data paying for participation?

There are some tables in Suppl. Mater., but it would be worth to discuss the difference between paid and unpaid (might it is not a problem of being (or not) paid but the demographic (e.g. age) structure is different in these two groups.

Due to the results obtained (selection was shown to be usually stronger on height or body shape than on penis size) I would suggest to change the title into for instance:

a.) Body height, shape and penis size - perceived attractiveness and assessment of fighting men ability, or

b.) The effect of height, body shape and penis size on perceived attractiveness and assessment of fighting ability in men.

I am not sure if Tab S4 is necessary. The results presented in this tab. are similar to Tab. 3

Reviewer #3:

I enjoyed reading this paper. It covers a topic that will be of broad interest, and the methods, analysis and writing are good. An advantage is the investigation of linear, quadratic, and correlational selection gradients, which is somewhat uncommon in human sexual selection research. Another advantage is the collection of both in-person/life-size data and online data. I have the following suggestions for strengthening the manuscript, particularly in regard to theoretical framing and interpreting the results:

1. Sexual ornament vs. status badge: Effects of flaccid penis length on perceptions of both attractiveness and fighting ability are interesting and novel. The authors state they "believe that the most important finding of our study is that penis size influences how males assess a rival's fighting ability." I agree that this is interesting, but to me, the most interesting finding is that the effect of penis size on attractiveness to women is approximately 5 times the magnitude of the effect on male perceptions of fighting ability. One reason that this finding is interesting and important is that it is the reverse relationship from other putative display traits in human males, such as beards (Neave & Shields, 2008) and deep voices (Aung et al., 2024), which more strongly influence perceptions of dominance and fighting ability among males than they increase attractiveness to females. In other words, whereas beards and deep voices appear to function primarily as badges of status (Aung et al., 2023; Grueter, Isler, & Dixson, 2015), penile length looks like a sexual ornament from the present data. A thorough review of sexual selection in human males that provides evidence that male secondary sex traits function primarily in contests is Puts, Carrier, and Rogers (2022). The present results suggest that penis size is an important exception to this overall pattern. In my 2010 EHB paper that the authors reference, I asked: "Are any of men's traits properly considered sexual ornaments (i.e., function primarily in mate attraction)? Men's penises are longer and thicker, both relatively and absolutely, than those of our closest relatives, chimpanzees and gorillas, and could have evolved to signal mate quality. Women report greater satisfaction with larger penises (Lever, Frederick, & Peplau, 2006), so penis size may affect a man's ability to stimulate orgasm in women (Miller, 2000). Female orgasm may boost sperm retention, facilitate sperm activation, and encourage additional copulations (reviewed in Puts & Dawood, 2006, see also Gallup et al., 2003). However, it has also been suggested that penis size may advertise vigor to other men (Diamond, 1997)." The present results thus provide the strongest evidence yet addressing this question. On this theme, I appreciated the authors' discussion of ancestral constraints on female mate choice, which is probably accurate; however, they might also note that the existence of strong female preferences in modern humans suggests that these preferences were functional ancestrally.

2. Primary functions in display vs. fighting success: It seems important to distinguish between traits that function more directly in winning fights, such as size and muscularity, and those that seem to function more in display, such as deep voices, and perhaps penis length. In the context of the present paper, it makes sense that males and females would have evolved sensitivity to the size and body shape of males because of what those traits indicate about a male's fighting ability, genetic quality, status, etc., but greater male size and strength evolved tens of millions of years ago, probably mainly in the service of winning fights directly, and perceptions evolved secondarily to that. This is different from penis size.

3. Primary function of penis size in display vs. sexual stimulation: The authors discuss copulatory sexual selection on male genitalia in their introduction and then note that precopulatory sexual selection on male genitalia is understudied; hence, the focus of the present paper. It might be worth returning in the discussion to the idea that both forms of sexual selection may have been important pressures on human penile length. Related to this and the idea that orgasm increases the probability of fertilization mentioned in the introduction, a thorough review of the evidence is Wheatley and Puts (2015).

4. Why does penis length influence perceptions of fighting ability? "One possible explanation why a larger penis might be an honest signal of fighting ability is that testosterone influences penile development at puberty, and higher levels of testosterone in adult males are associated with increased muscle mass, greater aggression, and higher competitive ability [42, 57-58." Agreed. Another reason may be that flaccid penis length responds to stress/anxiety by shortening as adrenaline redirects blood flow away from the penis. This response may function to decrease injury to the penis during fight or flight situations. Research subjects may be interpreting penis length partly as a state rather than a trait, with longer penises perceived as indicating confidence, status, and a relative absence of plausible threats.

5. Effects of perceivers' traits on their perceptions: The present finding of effects of own height on men's perceptions should be discussed in the context of a prior literature on this: (Watkins, Fraccaro, et al., 2010; Watkins, Jones, & DeBruine, 2010; Wolff & Puts, 2010). Watkins et al. found that less dominant and shorter men were more attentive to indicators of the dominance of other men, whereas Wolff & Puts found no relationship. I believe I've seen one or two other studies on this topic more recently as well. The finding of an effect of male perceiver's age on perceptions male body shape is similar to the finding of Aung et al. (2024) that low voice pitch more strongly influenced perception of fighting ability in younger men and perceptions of social status in older men. Like the present authors, Aung et al. interpreted their results in the context of age-related changes in the ways that male compete.

6. Typos/minor edits: "In humans, a few experimental studies have presented women with computer generate images…" Change to "generated" "…600 male and 200 female participants ranked figures that independently varied in three traits" Change to rated?

I hope these comments help interpret and explain the importance of their study. These are unique data that shed significant light on major questions in human sexual selection.

- Signed, David Puts

References

Aung, T., Hill, A. K., Hlay, J. K., Hess, C., Hess, M., Johnson, J., . . . Puts, D. (2024). Effects of Voice Pitch on Social Perceptions Vary With Relational Mobility and Homicide Rate. Psychological Science, 35(3), 250-262. doi:10.1177/09567976231222288

Aung, T., Hill, A. K., Pfefferle, D., McLester, E., Fuller, J., Lawrence, J. M., . . . Puts, D. A. (2023). Group size and mating system predict sex differences in vocal fundamental frequency in anthropoid primates. Nature communications, 14(1), 4069. doi:10.1038/s41467-023-39535-w

Grueter, C. C., Isler, K., & Dixson, B. J. (2015). Are badges of status adaptive in large complex primate groups? Evolution and Human Behavior, 36(5), 398-406.

Neave, N., & Shields, K. (2008). The effects of facial hair manipulation on female perceptions of attractiveness, masculinity, and dominance in male faces. Personality and Individual Differences, 45(5), 373-377. doi:10.1016/j.paid.2008.05.007

Puts, D., Carrier, D., & Rogers, A. R. (2022). Contest competition for mates and the evolution of human males. In D. M. Buss (Ed.), The Oxford handbook of human mating (pp. 317-377). Oxford, UK: Oxford University Press.

Watkins, C. D., Fraccaro, P. J., Smith, F. G., Vukovic, J., Feinberg, D. R., DeBruine, L. M., & Jones, B. C. (2010). Taller men are less sensitive to cues of dominance in other men. Behavioral Ecology, 21(5), 943-947. doi:10.1093/beheco/arq091

Watkins, C. D., Jones, B. C., & DeBruine, L. M. (2010). Individual differences in dominance perception: Dominant men are less sensitive to facial cues of male dominance. Personality and Individual Differences, 49(8), 967-971. doi:10.1016/j.paid.2010.08.006

Wheatley, J. R., & Puts, D. A. (2015). Evolutionary science of female orgasm. In T. K. Shackelford & R. D. Hansen (Eds.), The Evolution of Sexuality (pp. 123-148): Springer.

Wolff, S. E., & Puts, D. A. (2010). Vocal masculinity is a robust dominance signal in men. Behavioral Ecology and Sociobiology, 64(10), 1673-1683. doi:10.1007/s00265-010-0981-5

---

## [Decision Letter · Decision Letter 2]

19 Dec 2025

Dear Dr Aich,

Thank you for the submission of your revised Research Article "Experimental evidence that penis size, height and body shape influence assessment of male sexual attractiveness and fighting ability in humans" for publication in PLOS Biology. On behalf of my colleagues and the Academic Editor, Gail Patricelli, I'm pleased to say that we can in principle accept your manuscript for publication, provided you address any remaining formatting and reporting issues. These will be detailed in an email you should receive within 2-3 business days from our colleagues in the journal operations team; no action is required from you until then. Please note that we will not be able to formally accept your manuscript and schedule it for publication until you have completed any requested changes.

Sincerely, 

Roli Roberts

Senior Editor

PLOS Biology

rroberts@plos.org

REVIEWER'S COMMENTS:

Reviewer #3:

I thank the authors for their careful attention to my comments and congratulate them in an important contribution. I have no further suggestions.